# Linked symptom monitoring and depression treatment programmes for specialist cancer services: protocol for a mixed-methods implementation study

Marta Wanat,[1] Jane Walker,[1] Katy Burke,[1] Nick Sevdalis,[2] Alison Richardson,[3] Amy Mulick,[4] Chris Frost,[4] Michael Sharpe[1]

[1]Department of Psychiatry, Psychological Medicine Research, University of Oxford, Warneford Hospital, Oxford, UK
[2]Centre for Implementation Science, King's College London, London, UK
[3]Faculty of Health Sciences, University of Southampton, Southampton, UK
[4]Department of Medical Statistics, London School of Hygiene and Tropical Medicine, London, UK

**Correspondence to**
Dr Marta Wanat; marta.wanat@psych.ox.ac.uk

## ABSTRACT

**Introduction** There is growing awareness that cancer services need to address patients' well-being as well as treating their cancer. We developed systematic approaches to (1) monitoring patients' symptoms including depression using a 'Symptom Monitoring Service' and (2) providing treatment for those with major depression using a programme called 'Depression Care for People with Cancer'. Used together, these two programmes were found to be highly effective and cost-effective in clinical trials. The overall aims of this project are to: (1) study the process of introducing these programmes into routine clinical care in a large cancer service, (2) identify the challenges associated with implementation and how these are overcome, (3) determine their effectiveness in a routine non-research setting and (4) describe patients' and clinicians' experience of the programmes.

**Methods and analysis** This is a mixed-methods longitudinal implementation study. We will study the process of implementation in three phases (April 2016–December 2018): 'Pre-implementation' (setting up of the new programmes), 'Early Implementation' (implementation of the programmes in a small number of clinics) and 'Implementation and Maintenance' (implementation in the majority of clinics). We will use the following methods of data collection: (1) contemporaneous logs of the implementation process, (2) interviews with healthcare professionals and managers, (3) interviews with patients and (4) routinely collected clinical data.

**Ethics and dissemination** The study has been reviewed by a joint committee of Oxford University Hospitals National Health Service Foundation Trust Research and Development Department and the University of Oxford's Clinical Trials and Research Governance Department and judged to be service evaluation, not requiring ethics committee approval. The findings of this study will guide the scaling up implementation of the programmes across the UK and will enable us to construct an implementation toolkit. We will disseminate our findings in publications and at relevant national and international conferences.

## INTRODUCTION

Worldwide, 14.1 million people are diagnosed with cancer each year.[1] While there have been significant advances in anticancer treatments,

### Strengths and limitations of this study

► The mixed-methods design of this study will produce a comprehensive summary of the process of and barriers and facilitators to implementation.

► The implementation of our depression screening and treatment programmes was explicitly advocated in the 2015 National Health Service Cancer Taskforce report, and the findings of this study will inform the process of implementation in other centres.

► The timeline of the study may be affected by the pace of implementation.

► The study findings may not be readily generalisable to other healthcare systems.

► There may be difficulties recruiting patients who do not use Symptom Monitoring Service or Depression Care for People with Cancer, which may limit our understanding of these patients' views towards the services.

leading to a longer life expectancy for many patients, there remains a pressing need to also improve patients' quality of life by better management of common cancer-related symptoms such as depression, pain and fatigue.[2]

As part of a research programme, we developed two clinical programmes which aimed to improve the quality of life of patients attending cancer services. The first was a systematic approach to the assessment of symptoms, including depression, called the 'Symptom Monitoring Service' (SMS). The second was a system of care for those patients with cancer identified as having depression, called 'Depression Care for People with Cancer' (DCPC).[3] We prioritised the treatment of depression as its management is especially challenging for cancer services. Our research found that, used together, SMS and DCPC are highly effective and cost-effective in improving the lives of those patients with cancer who suffer from comorbid depression.[4–8]

BMJ

## Symptom Monitoring Service

The SMS provides a systematic way of monitoring patients' symptoms, including depression, every time they attend the clinic. Approximately 10% of cancer outpatients have comorbid major depression.[9] Depression substantially reduces patients' quality of life and ability to cope with anticancer treatments but often goes undetected and untreated.[9–11] Patients using the SMS are assisted to complete brief questionnaires about relevant symptoms using touchscreen computers while waiting for their appointment. These questionnaires include symptoms relevant to all clinics (eg, depressive symptoms and pain) as well as symptoms relevant to specific clinics (eg, coughing up blood in the lung cancer clinic). The results of these questionnaires are available to the patient's clinician (eg, oncologist or cancer nurse specialist) before the appointment. The clinician can use this information to: (1) identify patients with severe symptoms; (2) detect changes in patients' symptoms over time; (3) determine which symptoms require further assessment; (4) ensure that patients receive appropriate treatment for their symptoms. To identify patients with major depression (the diagnostic term for depression which requires treatment), those with a high score on the relevant symptom questionnaire are offered a telephone-delivered diagnostic interview.[12]

These semistructured clinical interviews are based on the major depression section of the Structured Clinical Interview for the Diagnostic and Statistical Manual of Mental Disorders.[13] Telephone interviews are used because these are more feasible than attempting to achieve interviews in the clinic, are convenient and acceptable to patients and have been found to have good agreement with face-to-face interviews.[14 15] The SMS is delivered by a team of symptom monitoring assistants, trained and supervised by consultation-liaison psychiatrists; further details are given in our previous publication.[16] Patients identified as having a diagnosis of major depression are offered participation in the linked DCPC treatment programme.

## Depression Care for People with Cancer

DCPC is a manualised, systematic, multicomponent, team-delivered treatment programme for patients with cancer and major depression that is based on the collaborative care model.[3 17] The treatment team comprises care managers (specially trained cancer nurses) and consultation-liaison psychiatrists who work closely with the patient's general practitioner (GP) and oncologist. The care managers establish a therapeutic relationship with the patient, provide education about depression, deliver brief evidence-based psychological treatments (behavioural activation and problem solving therapy) and systematically monitor the patient's progress.[18 19] The psychiatrists supervise treatment with the aim of achieving and then maintaining specified treatment targets, advise GPs and oncologists on the prescription of antidepressant medication and also provide direct consultations to those patients who are not improving.

## The process of implementing a new service

Implementation of these systems has been explicitly advocated in the 2015 National Health Service (NHS) Cancer Taskforce report.[20] However, it is well known that the translation from delivering services as part of research programmes to delivering them in routine clinical practice can be difficult and may lead to innovative services failing to realise their promise.[21]

### Aim

Our aim is to study the implementation of SMS and DCPC into the routine clinical care provided by a large cancer service in Oxford, UK, with a view to learning how best to facilitate the introduction of these programmes into other cancer services. We will: (1) describe the process of implementation, (2) identify the challenges in implementation and how these are overcome, (3) determine the programmes' effectiveness in clinical practice and (4) describe patients' and clinicians' experience of them. Detailed aims are described in table 1.

## METHODS AND ANALYSIS

### Study design

We will use a mixed-methods design to longitudinally study the process of introducing SMS and DCPC into the Oxford Cancer Centre. The Oxford Cancer Centre is a large cancer service, which is part of Oxford University Hospitals NHS Foundation Trust. It receives approximately 19 000 new referrals per year, has 58 consultant oncologists and comprises inpatient wards, operating theatres, a day hospital and outpatient departments.

The study will focus on three phases: 'Pre-implementation' (setting up of the new programmes), 'Early Implementation' (implementation in a small number of clinics) and 'Implementation and Maintenance' (implementation in the majority of clinics). Our specific aims are described in detail in table 1. In the pre-implementation phase, we will study the process of setting up the new programmes, the challenges encountered during set-up and the ways these are addressed. In the early implementation and implementation and maintenance phases, we will describe the challenges encountered and how they are addressed, the effectiveness of the programmes in clinical practice and the experiences of both the patients who have access to SMS and DCPC and of the healthcare professionals involved in them.

We will use the following methods of data collection: (1) contemporaneous logs of the implementation process, (2) interviews with healthcare professionals and managers, (3) interviews with patients and (4) routine clinical data. Data collection in the Early Implementation and Implementation and Maintenance phases will be guided by the RE-AIM (Reach, Effectiveness, Adoption, Implementation, Maintenance) framework.[22 23]

**Table 1** Study aims related to each phase of implementation

| Phase of implementation | Study aim | Relevant RE-AIM components | Data collection methods |
|---|---|---|---|
| Pre-implementation | Describe the timeline and process of setting up the new programmes in the cancer service | N/A | Contemporaneous log |
| | Describe the challenges of setting up the programmes in the cancer service and the ways these are addressed including the experiences of the health professionals and managers involved in set-up | N/A | Contemporaneous log, interviews with healthcare professionals and managers |
| Early implementation | Describe the timeline and process of introducing SMS and DCPC in a small number of clinics | Implementation, Adoption | Contemporaneous log |
| | Describe the challenges associated with day-to-day running and how these are addressed | Implementation, Adoption, Effectiveness | Contemporaneous log, interviews with healthcare professionals |
| | Describe the experiences of healthcare professionals involved in SMS and DCPC | Implementation, Adoption, Effectiveness, Reach | Interviews with healthcare professionals |
| | Describe patients' experience of SMS and DCPC and how these affect their use of them | Effectiveness, Reach | Interviews with patients |
| | Measure the clinical effectiveness of DCPC by comparing patient outcomes with those achieved in clinical trials | Effectiveness | Routine clinical data |
| | Measure the proportion and characteristics of the target population who have access to and who use SMS and DCPC | Reach | Routine clinical data |

Continued

**Table 1** Continued

| Phase of implementation | Study aim | Relevant RE-AIM components | Data collection methods |
|---|---|---|---|
| Implementation and maintenance | Describe the timeline and process of the introduction of SMS and DCPC in the majority of clinics | Adoption, Maintenance | Contemporaneous log |
| | Describe the challenges associated with implementation and long-term maintenance and how these are addressed | Implementation, Maintenance | Contemporaneous log, interviews with healthcare professionals |
| | Describe the challenges of delivering SMS and DCPC as intended and adaptations made to the programmes | Implementation, Maintenance | Contemporaneous log, interviews with healthcare professionals |
| | Describe the experiences of healthcare professionals involved in SMS and DCPC | Adoption, Effectiveness | Interviews with healthcare professionals |
| | Describe patients' experience of SMS and DCPC and how these affect their use of them | Effectiveness, Reach | Interviews with patients |
| | Measure the effectiveness of DCPC over time by comparing patient outcomes with those achieved in clinical trials | Effectiveness, Maintenance | Routine clinical data |
| | Determine the extent to which SMS and DCPC are delivered as intended | Implementation | Routine clinical data |
| | Measure the proportion and characteristics of the target population who have access to and who use SMS & DCPC over time | Reach | Routine clinical data |
| | Measure the cost of delivering SMS and DCPC | Implementation | Routine clinical data |

DCPC, Depression Care for People with Cancer; N/A, not applicable; RE-AIM, Reach, Effectiveness, Adoption, Implementation, Maintenance; SMS, Symptom Monitoring Service.

## Preimplementation phase
### Contemporaneous log

A contemporaneous events log will be used by the study team to record any activities and events related to setting up the new programmes in the cancer service and will include regular correspondence from staff involved in setting up the service.[24] Sample activities and events to be included are recruitment and training events, and meetings with clinical managers and information technology experts. Each entry to the log will include information on the date of the action or event, the people involved and its effects on the delivery of the programmes.

### Interviews with healthcare professionals and managers

We will recruit and interview 10–15 healthcare professionals and managers who are involved in setting up SMS and DCPC in the cancer service. Based on similar studies, we anticipate this sample size will be sufficient to reach data saturation.[25] Interviews will take approximately 30 min, will be audio-recorded and transcribed verbatim and will be conducted at the participants' preferred location or by telephone. Interviews will be semistructured following a topic guide which will focus on participants' views of the process of setting up the programmes. They will be conducted by a trained member of the study team with previous experience of interviewing healthcare professionals.

### Analysis

A timeline of events that occur during set-up will be drawn using the contemporaneous log. Data from the log will then be analysed using content analysis, which will involve data familiarisation and coding. Codes will be developed focusing on the process of setting up the programmes, the associated challenges and the ways that these are addressed. The codes developed from the contemporaneous log will be applied to the transcripts of interviews with healthcare professionals and managers and will be further refined. These codes will be then sorted into subcategories and categories.

## Early implementation and implementation and maintenance phases
### Contemporaneous log

We will record and use data in a contemporaneous events log in the same away as in the pre-implementation phase. This will allow us to describe the time taken for each step of implementation.

### Interviews with healthcare professionals and managers

We will recruit 20–30 healthcare professionals and managers during the Early Implementation and Implementation and Maintenance phases. These will include psychiatrists and care managers involved in the delivery of the SMS and DCPC, and oncologists, cancer nurse specialists and GPs who make up the wider collaborative care team. Based on similar studies, we anticipate this sample size will be sufficient to reach data saturation.[25] Semistructured Interviews will take approximately 30 min, will be audio-recorded and transcribed verbatim and will be conducted at the participant's preferred location or by telephone. Interviews will be conducted by a trained member of the study team with previous experience of interviewing healthcare professionals. The interview topic guide will focus on the participants' experience of the new programmes.

### Interviews with patients

We will recruit and interview 40–60 patients during the Early Implementation and Implementation and Maintenance phases. These will include patients who do not use the first stage of the SMS service (n=10–15), who use the SMS service (n=10–15), who do not use the DCPC treatment programme (n=10–15) and who use the DCPC treatment programme (n=10–15). We will exclude patients who are unable to provide informed consent to participate or do not understand and speak English sufficiently to participate in a qualitative interview. Interviews will take between 30 and 60 min, will be audio-recorded and transcribed verbatim and will be conducted at the participant's preferred location or by telephone. Interviews will be conducted by a trained member of the study team with previous experience of interviewing patients with cancer. The interview topic guide will focus on patients' understanding and views of the SMS and DCPC.

### Routine clinical data

We will obtain the following data from the cancer centre: the number and characteristics of clinics with and without the new programmes over time, the proportion of patients who do and do not receive each part of the programmes and their characteristics, the staff and other resources required and costs of delivering SMS and DCPC, adherence to the SMS and DCPC manuals including diagnostic accuracy, outcomes of patients who receive the programmes and description of any relevant adverse clinical events.

### Analysis

Data recorded in the contemporaneous log related to the introduction of SMS and DCPC in clinics will be used to draw a timeline across the Early Implementation and Implementation and Maintenance phases. Then, data from the contemporaneous log will be analysed using content analysis, which will involve data familiarisation and coding. Codes will be developed focusing on: the process of implementation, challenges associated with day-to-day running of SMS and DCPC, challenges associated with implementation and long-term maintenance, challenges associated with delivering SMS and DCPC as intended and the ways that these challenges are addressed. Codes developed from the contemporaneous log will be applied to selected relevant excerpts from the transcripts of qualitative interviews with healthcare professionals. These codes will be further refined and then sorted into subcategories and categories.

Data generated from the qualitative interviews with patients and with healthcare professionals involved in SMS and DCPC will be analysed using thematic analysis to describe their experiences.[26] The stages of the analysis will include data familiarisation, coding and development of themes. Data collection and analysis will take place simultaneously to allow the exploration of themes from the initial interviews in subsequent ones. In order to minimise subjective biases, the coding will be discussed among members of the study team.

We will analyse the routine clinical data using descriptive statistics in order to summarise: the effectiveness of DCPC over time (including benchmarking against the patient outcomes in our clinical trials), the proportion and characteristics of the target population who have access to and who use SMS and DCPC over time, the fidelity of the delivery of SMS and DCPC, adherence of patients with each component of the DCPC and the cost of the new services. We will use data from qualitative interviews to understand these results, for example, how challenges in the adherence to the protocol may influence the effectiveness of the service.

### Ethics and dissemination

The study has been reviewed by a joint committee of Oxford University Hospitals NHS Foundation Trust Research and Development Department and the University of Oxford's Clinical Trials and Research Governance Department and has been judged to be service evaluation, not requiring ethics committee approval. The study has been registered with the Oxford University Hospitals NHS Foundation Trust service evaluation and clinical audit team. The results of the study will be disseminated via presentations at local, national and international conferences, peer-reviewed journals and workshops with stakeholders. The implementation of SMS and DCPC in UK cancer centres is explicitly advocated in the 2015 NHS Cancer Taskforce report.[10] The findings from the study will be crucial for scaling up implementation and will form part of a toolkit for doing this.

## DISCUSSION

Our overall aim is to study the process of setting up and introducing two new linked clinical programmes (SMS and DCPC) in a large UK cancer service, to describe the challenges associated with their implementation and how these are addressed, to measure their effectiveness in clinical practice and to describe patients' and clinicians' experience of using them. While these programmes have already been successfully implemented in the context of a large research programme and have been found to be effective and cost-effective in clinical trials, this is the first time that they have been implemented in a non-research setting.

The Early Implementation and Implementation and Maintenance phases of our study will use the RE-AIM framework to guide data collection, analysis and reporting of findings. The authors of this framework have recently highlighted the importance of collecting data in relation to all components of RE-AIM framework and especially qualitative data about patients not using the service.[23] Our study follows both of these recommendations.

We anticipate a number of challenges in conducting this research. First, the timeline of the study will be affected by the pace of implementation. We have divided the study into three phases to minimise delays to data collection. Second, there may be difficulties in recruiting busy healthcare professionals and managers for interviews. We have mitigated this risk by working closely with healthcare professionals and managers to identify research questions which are of interest to them.

Thirdly, this study involves multiple methods of data collection. We will make sure that we integrate qualitative and quantitative findings to further our understanding of the challenges associated with the implementation of the clinical programmes. Finally, we will also make sure that the research team works sufficiently close with the implementation team to be able to gather data while remaining independent enough to portray the problems accurately.[27]

This study will provide us with robust new knowledge, essential for service planning, about the process of implementing these two linked programmes and the challenges to implementation success. We will also find out what patients and clinicians think of SMS and DCPC and how they can be further improved. The findings of this study will be crucial for scaling up implementation of the programmes across the UK and will enable us to construct a toolkit for this purpose. More widely, the study findings will further our understanding of the mechanisms involved in implementing new systems of care within an established clinical service.

**Contributors** MW: study design and conduct; manuscript writing. JW: study conception, obtained funding, study design and conduct, manuscript writing. KB, NS, AR, AM, CF: advised on the study design, manuscript review. MS: study conception, obtained funding, study design and conduct, manuscript writing.

**Funding** This study is funded by the National Institute for Health Research (NIHR) Collaboration for Leadership in Applied Health Research and Care Oxford at Oxford Health NHS Foundation Trust. The clinical programmes are funded by Oxford University Hospitals NHS Foundation Trust in collaboration with Macmillan Cancer Support charity. NS' research is supported by the NIHR Collaboration for Leadership in Applied Health Research and Care South London at King's College Hospital NHS Foundation Trust. NS is a member of King's Improvement Science, which is part of the NIHR CLAHRC South London and comprises a specialist team of improvement scientists and senior researchers based at King's College London. Its work is funded by King's Health Partners (Guy's and St Thomas' NHS Foundation Trust, King's College Hospital NHS Foundation Trust, King's College London and South London and Maudsley NHS Foundation Trust), Guy's and St Thomas' Charity, the Maudsley Charity and the Health Foundation. AR's research is supported by the NIHR Collaboration for Leadership in Applied Health Research and Care Wessex at the University Hospital Southampton NHS Foundation Trust.

**Disclaimer** The views expressed are those of the authors and not necessarily those of the NHS, the NIHR or the Department of Health.

**Competing interests** NS is the director of London Training & Safety Solutions, which consults on team training, human factors and patient safety.

**Ethics approval** The study has been reviewed by a joint committee of Oxford University Hospitals NHS Foundation Trust Research and Development Department and the University of Oxford's Clinical Trials and Research Governance Department.

**Provenance and peer review** Not commissioned; externally peer reviewed.

**Data sharing statement** We do not plan to share data obtained during this study.

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
