## [Reviewer comments · BMJ Open]

ARTICLE DETAILS

TITLE (PROVISIONAL)	Linked symptom monitoring and depression treatment programmes for specialist cancer services: Protocol for a mixed-methods implementation study.
AUTHORS	Wanat, Marta; Walker, J.; Burke, Katy; Sevdalis, Nick; Richardson, Alison; Mulick, Amy; Frost, Chris; Sharpe, Michael

VERSION 1 - REVIEW

REVIEWER	Sean Marks Medical College of Wisconsin Milwaukee, Wisconsin United States of America
REVIEW RETURNED	07-Apr-2017

GENERAL COMMENTS	Overall this is a great research proposal. In general, I found it to be well written and the research questions to be clinically practical and useful. I foresee this being a very worthwhile investigation for not just clinicians in the UK, but clinicians in many other countries as well. So I look forward to the results. I think there are some opportunities for improvement, however, which I'll outline below. 1. The title. The word introducing seems misplaced and I'd suggest removing it. I'm also wondering if there was a way for the author's to be a little more clear on what is meant by "linked symptom monitoring" in the title. Sounds a bit too much like jargon and I worry that because of that they may lose interested readers. Perhaps something along the lines of "The description and feasibility of a systematic symptom monitoring and treatment programme for depression in a large cancer service: a mixed-methods implementation study protocol." 2. Strengths and Limitations section: the first bullet point seems a bit vague and I think there's an opportunity to state this more clearly by highlighting that the mixed-methods design should offer a more complete view of feasibility of this programme via utilizing both qualitative and quantitative data. There are two other limitations, which may be worth noting in this section. a) Their treatment programme may not be generalizable or as feasible in other health-care systems, specifically health systems like the US which operate on more of a pay-for-service model. From what I can glean much of the screening may require a licensed clinician who can properly make the diagnosis of depression to spend significant non-billable time performing telephone interviews. This could be difficult to reproduce for many health care centers. b) It attempts to investigate the feasibility and satisfaction from a patient and clinician perspective regarding the implementation of an
--

evidence-based screening and treatment programme for depression in advance cancer patients in a more "real-world" setting than described in the initial trial. However, the authors are receiving funding for this research and the research is occurring in a controlled environment. Hence, by its nature there will be difficulty in applying these results to a non-research setting.

c) I don't believe financial cost information is being collected nor would that cost be generalizable in different health care systems. Perhaps that therefore should be stated in this section.

Introduction: overall very well written. I did have several questions namely about the SMS. Forgive me if these have been addressed in the previous study, but in case other readers may have similar questions, allow me to proffer them.

a) For the SMS, what happens if the clinician performing the screening encounters suicidal ideation or other signs of a mental health crisis? Is there a clear protocol in place for the clinician to follow that keeps the patient safety in mind?

b) Is this screening billable? For many health systems it may not be financially feasible to have a clinical psychologist or licensed physician who could make the diagnosis of depression available to devote the time via telephone to do this screening if it's not billable.

c) Is the screening a structured interview by someone credentialed to make a diagnosis of major depression and distinguish it from potential common diagnoses which may mimic depression in advance cancer -- e.g. hypoactive delirium, grief, adjustment reaction, uncontrolled physical symptoms like pain, substance abuse, personality disorder etc? Could this structured interview be easily recreated? Are they using a specific depression screening instrument?

Seeing that depression can be quite tricky to diagnose in a seriously ill patient population, these questions seem relevant to me if the prevailing question being tested is, is this screening and treatment programme feasible in a non-research setting.

3. Aim: I like how the four principle aims are generally categorized into four parts. But I think there is an opportunity for each of these 4 aims to be more specified. For example, with aim #1: what are they trying to understand about the implementation process. How long it takes? How many resources are required to implement it? Are they merely trying to describe the process so it could be more reproducible? Regarding aim #3: I would suggest they find different phrasing than "real world" to describe their intentions. As alluded to previously, this is a colloquial phrase which does not seem entirely accurate for the reasons mentioned before.

4. Study design: Again, I like how this is a mix-methods design and it is longitudinal. This will be a major strength of the study along with the large number of patients. However, I would like to see more specifics on how they are going to test their aims in both a quantitative and qualitative sense. In particular, what specific quantitative outcomes are they going to measure? Length of implementation? Clinical time each step requires? Cost? Number of different clinicians needed? Staff resources needed? Was the diagnosis of depression made in a timely fashion? Accurate fashion? Were there unintended consequences -- delays in diagnosis, improper clinical decisions regarding patient statements reflecting acute mental health crises?

Regarding the qualitative data: what type of healthcare professional will be carrying out the interviews? Will they be standardized? How

	will the themes in the discussions be analyzed/coded in a qualitative sense? 5. Discussion section: The last sentence of the first paragraph seems to contradict itself by stating, "this is the first time that they have been implemented in a non-research setting." Seems like a wrong choice of words seeing that this is itself a research study. I think if the authors can more concretely flesh out the answers to these questions in the manuscript, their investigation and results will be more reliable, generalizable, and widely shared. This will benefit patients, readers, and them as authors.
--	--

REVIEWER	Harm van Marwijk University of Manchester, UK
REVIEW RETURNED	23-Apr-2017

GENERAL COMMENTS	Nice project, it is a bit difficult to assess the size as it focuses on implementation. Is a theory used, such as Normalisation Process Theory? The methods are fine but could perhaps be explained in a bit more detail. Most implementation trials now have a particularly close look at the link between implementation and patient outcomes. See for instance, van Beljouw IMJ, Laurant MGH, Heerings M, Stek ML, van Marwijk HWJ, van Exel E: Implementing an outreaching, preference-led stepped care intervention programme to reduce late life depressive symptoms: results of a mixed-methods study. Implement Sci 2014 Jan;9:107.
--

VERSION 1 – AUTHOR RESPONSE

Reviewer	Specific comment to be addressed	Response to comments and changes made
1	Overall this is a great research proposal. In general, I found it to be well written and the research questions to be clinically practical and useful. I foresee this being a very worthwhile investigation for not just clinicians in the UK, but clinicians in many other countries as well. So I look forward to the results. I think there are some opportunities for improvement, however, which I'll outline below.	Thank you very much for this positive feedback.
1	The title. The word introducing seems misplaced and I'd suggest removing it. I'm also wondering if there was a way for the author's to be a little more clear on what is meant by "linked symptom monitoring" in the title. Sounds a bit too much like jargon and I worry that because of that they may lose interested readers. Perhaps something along the lines of "The description and	Title has been changed to "Linked symptom monitoring and depression treatment programmes for specialist cancer services: Protocol for a mixed-methods implementation study".

	feasibility of a systematic symptom monitoring and treatment programme for depression in a large cancer service: a mixed-methods implementation study protocol."	
1	Strengths and Limitations section: the first bullet point seems a bit vague and I think there's an opportunity to state this more clearly by highlighting that the mixed-methods design should offer a more complete view of feasibility of this programme via utilizing both qualitative and quantitative data. There are two other limitations, which may be worth noting in this section (Strength and limitations). a) Their treatment programme may not be generalizable or as feasible in other health-care systems, specifically health systems like the US which operate on more of a pay-for-service model. From what I can glean much of the screening may require a licensed clinician who can properly make the diagnosis of depression to spend significant non-billable time performing telephone interviews. This could be difficult to reproduce for many health care centers. b) It attempts to investigate the feasibility and satisfaction from a patient and clinician perspective regarding the implementation of an evidence-based screening and treatment programme for depression in advance cancer patients in a more "real-world" setting than described in the initial trial. However, the authors are receiving funding for this research and the research is occurring in a controlled environment. Hence, by its nature there will be difficulty in applying these results to a non-research setting. c) I don't believe financial cost information is being collected nor would that cost be generalizable in different health care systems. Perhaps that therefore should be stated in this section.	We have updated the strengths and limitations to address these comments. We have also clarified in the Introduction that the SMS is delivered by a team of symptom monitoring assistants, trained and supervised by consultation-liaison psychiatrists and have provided a reference to a previous publication describing the programme in more detail. We have clarified that the implementation is not occurring in a controlled research environment but rather in usual clinical practice – we have now specified in the funding section that funding for the clinical programmes comes from the Oxford University Hospitals NHS Foundation Trust in collaboration with Macmillan Cancer Support charity. The research study is to describe the implementation and is separately funded. The study will include collection of cost information (please see Methods). We will provide detailed descriptions of the costs incurred, which will allow readers to identify costs applicable to their own healthcare systems.

1	Introduction: overall very well written. I did have several questions namely about the SMS. Forgive me if these have been addressed in the previous study, but in case other readers may have similar questions, allow me to proffer them. a) For the SMS, what happens if the clinician performing the screening encounters suicidal ideation or other signs of a mental health crisis? Is there a clear protocol in place for the clinician to follow that keeps the patient safety in mind? b) Is this screening billable? For many health systems it may not be financially feasible to have a clinical psychologist or licensed physician who could make the diagnosis of depression available to devote the time via telephone to do this screening if it's not billable. c) Is the screening a structured interview by someone credentialed to make a diagnosis of major depression and distinguish it from potential common diagnoses which may mimic depression in advance cancer -- e.g. hypoactive delirium, grief, adjustment reaction, uncontrolled physical symptoms like pain, substance abuse, personality disorder etc? Could this structured interview be easily recreated? Are they using a specific depression screening instrument?	We appreciate the reviewer's consideration of the practical details of the SMS. We have added brief text to the Introduction and have referred readers to a comprehensive description of the SMS which we have recently published. The first stage of SMS is delivered by non-clinical SMS assistants, who help patients to complete the questionnaires. The second stage telephone interviews are based on the depression section of the Structured Clinical Interview for DSM. These are conducted by nurses, allied health professionals or non-clinical SMS assistants depending on the local service requirements and resources. Consultation-liaison psychiatrists train these staff and supervise them closely as well as being available to manage urgent situations. The billing arrangements will depend on how the SMS is implemented in different cancer services and the local arrangements for billing. We will provide information on the resources used in the Oxford service so that other providers can determine the optimal way of implementing SMS locally.
1	Aim: I like how the four principle aims are generally categorized into four parts. But I think there is an opportunity for each of these 4 aims to be more specified. For example, with aim #1: what are they trying to understand about the implementation process. How long it takes? How many resources are required to implement it? Are they merely trying to describe the process so it could be	We have amended the overall aims and show the detailed aims in a table. The term 'real-world' has been removed.

	more reproducible? Regarding aim #3: I would suggest they find different phrasing than "real world" to describe their intentions. As alluded to previously, this is a colloquial phrase which does not seem entirely accurate for the reasons mentioned before.	
1	Study design: Again, I like how this is a mix-methods design and it is longitudinal. This will be a major strength of the study along with the large number of patients. However, I would like to see more specifics on how they are going to test their aims in both a quantitative and qualitative sense. In particular, what specific quantitative outcomes are they going to measure? Length of implementation? Clinical time each step requires? Cost? Number of different clinicians needed? Staff resources needed? Was the diagnosis of depression made in a timely fashion? Accurate fashion? Were there unintended consequences - - delays in diagnosis, improper clinical decisions regarding patient statements reflecting acute mental health crises? Regarding the qualitative data: what type of healthcare professional will be carrying out the interviews? Will they be standardized? How will the themes in the discussions be analyzed/coded in a qualitative sense?	We have added more details to the Methods section to address these comments.
1	Discussion section: The last sentence of the first paragraph seems to contradict itself by stating, "this is the first time that they have been implemented in a non-research setting." Seems like a wrong choice of words seeing that this is itself a research study	This is not correct. The implementation itself is not a research study – the research is the study of the clinical implementation. We have now clarified this.
2	Nice project, it is a bit difficult to assess the size as it focuses on implementation. Is a theory used, such as Normalisation Process Theory? The methods are fine but could perhaps be explained in a bit more detail. Most implementation trials now have a particularly close look at the link between implementation and patient outcomes. See for instance, van Beljouw IMJ, Laurant MGH, Heerings	Thank you for this positive feedback. The study is guided by the RE-AIM framework. We have added further details in the methods section to clarify that we will collect data on both the implementation of the programmes, including fidelity in their delivery, and on patient outcomes which will be benchmarked

	M, Stek ML, van Marwijk HWJ, van Exel E: Implementing an outreaching, preference-led stepped care intervention programme to reduce late life depressive symptoms: results of a mixed-methods study. Implement Sci 2014 Jan;9:107.	against clinical trial data.
--	---	------------------------------

VERSION 2 – REVIEW

REVIEWER	Sean Marks MD Medical College of Wisconsin United States of America
REVIEW RETURNED	24-May-2017

GENERAL COMMENTS	Thank for submitting this revised version of your study proposal. I believe you have successfully addressed all of my comments.
---

REVIEWER	Harm van Marwijk University of Manchester, United Kingdom
REVIEW RETURNED	24-May-2017

GENERAL COMMENTS	Seems fine but I could not find the table with changes?
---

VERSION 2 – AUTHOR RESPONSE

Thank you for your response to our submission of this manuscript.
 We have addressed the editorial and reviewer 1 comments. We believe that the second reviewer's comment may be intended for a different manuscript.
 We hope that the paper is now suitable for publication in BMJ Open and look forward to hearing from you.